# Novel Quercetin Derivative of 3,7-Dioleylquercetin Shows Less Toxicity and Highly Potent Tyrosinase Inhibition Activity

**DOI:** 10.3390/ijms22084264

**Published:** 2021-04-20

**Authors:** Moon-Hee Choi, Seung-Hwa Yang, Da-Song Kim, Nam Doo Kim, Hyun-Jae Shin, Kechun Liu

**Affiliations:** 1Department of Chemical Engineering, Graduate School of Chosun University, Gwangju 61452, Korea; aamoony1222@naver.com (M.-H.C.); sh556@daum.net (S.-H.Y.); dasong1214@daum.net (D.-S.K.); 2VORONOI BIO Inc., Incheon 21984, Korea; namdoo@voronoibio.com; 3Biology Institute, Qilu University of Technology (Shandong Academy of Sciences), Jinan 250103, China; hliukch@sdas.org

**Keywords:** quercetin derivatives, 3,7-dioleylquercetin, tyrosinase inhibitor, anti-melanogenesis, zebrafish, enzyme kinetics, skin whitening agent

## Abstract

Quercetin is a well-known plant flavonol and antioxidant; however, there has been some debate regarding the efficacy and safety of native quercetin as a skin-whitening agent via tyrosinase inhibition. Several researchers have synthesized quercetin derivatives as low-toxicity antioxidants and whitening agents. However, no suitable quercetin derivatives have been reported to date. In this study, a novel quercetin derivative was synthesized by the S_N_2 reaction using quercetin and oleyl bromide. The relationship between the structures and activities of quercetin derivatives as anti-melanogenic agents was assessed using in vitro enzyme kinetics, molecular docking, and quenching studies; cell line experiments; and in vivo zebrafish model studies. Novel 3,7-dioleylquercetin (OQ) exhibited a low cytotoxic concentration level at >100 µg/mL (125 µM), which is five times less toxic than native quercetin. The inhibition mechanism showed that OQ is a competitive inhibitor, similar to native quercetin. Expression of tyrosinase, tyrosinase-related protein 1 (TRP-1) and tyrosinase-related protein 2 (TRP-2), and microphthalmia-associated transcription factor was inhibited in B16F10 melanoma cell lines. mRNA transcription levels of tyrosinase, TRP-1, and TRP-2 decreased in a dose-dependent manner. Melanin formation was confirmed in the zebrafish model using quercetin derivatives. Therefore, OQ might be a valuable asset for the development of novel skin-whitening agents.

## 1. Introduction

Melanin is a naturally occurring pigment that produces eyes, hair, and skin color. An amino acid, tyrosine, is required to support the production of melanin. Decreased melanin production may have a protective effect on the skin. The growing demand for whitening agents in cosmetic products has had a significant influence on anti-melanogenesis research. The inhibition of melanin biosynthesis (hypopigmentation or anti-melanogenesis) in the skin plays a critical role in protecting against various skin problems, such as spots, melasma, and freckles. Excessive production and accumulation of melanin (hyperpigmentation and melanogenesis) may cause serious skin diseases, such as skin cancer [1]. Therefore, many scientists have been trying to find a safe and effective whitening agent for cosmetic and therapeutic applications. To date, many potent tyrosinase inhibitors have been isolated, developed, and used in various skincare products, including ascorbic acid, kojic acid, arbutin, hydroquinone, and quercetin [2,3]. Among them, arbutin has been used as a positive control in numerous whitening studies. It is a hydroquinone glycoside with two isoforms, 4-hydroxypheyl-α-glucopyranoside and 4-hydroxylpheyl-β-glucopyranoside. Numerous studies have shown that arbutin is as effective as hydroquinone but less toxic [4]. The alpha isomer has the greatest inhibitory activity against mammalian tyrosinases [5].

As various natural resources show antioxidant activity, phenolic compounds are frequently used as cosmetic ingredients. In particular, quercetin (3,3′,4′,5,7-pentahydroxyl-flavone), a model polyphenol compound, is well known as a strong antioxidant and anti-whitening agent [3,6,7,8]. Quercetin has been widely used as a cosmetic material because it is present in large quantities in many plants. Quercetin is a potent tyrosinase inhibitor, a melanogenesis inhibitor, in several cell lines, such as mouse B16 melanoma, and an antioxidant and anticancer agent. However, there have been some controversies regarding the anti-melanogenesis effect based on the origin, concentration, and assay methods. In addition to native quercetin, several studies on the activity of plant extracts containing quercetin derivatives have been conducted to reduce the toxicity and inhibit tyrosinase activity. Quercetin 4′-O-β-D-glucopyranoside and its extract showed potent tyrosinase inhibitory activity [9,10]. Chao et al. [11] confirmed that the whitening effect was increased by attaching galactose, rhamnose, and xylose to the 3-OH position of quercetin. In addition, Taira et al. [12] verified the whitening effect and toxicity of quercetin-glucose-rhamnose derivatives. Concerning an artificial quercetin derivative, several unsaturated fatty acid-quercetin derivatives were synthesized; however, they showed no superior toxicity or kinetic constants than native quercetin. To achieve the goal of less toxicity and high inhibition activity of tyrosinase, facile synthesis of novel quercetin derivatives is required.

In the present study, we first prepared and purified novel 3,7-dioleylquercetin (OQ) by simple S_N_2 reaction with quercetin and fatty acid bromide, and then investigated their cytotoxicity and anti-melanogenesis activities. In addition, to determine whether the quercetin derivative is suitable as an anti-melanogenic agent, a toxicity test of quercetin and its derivatives was performed using cell lines and an in vivo zebrafish model. To verify the whitening effect theoretically, enzyme kinetics and docking and quenching studies were conducted.

## 2. Results

### 2.1. Synthesis and Identification of Novel Compounds

Synthesis was performed by modifying the method of Kato et al. [13]. The synthesis of OQ was undertaken via the S_N_2 reaction using quercetin and oleyl bromide (Scheme 1). Generally, the reaction occurs simultaneously at positions 3, 7, and 4′ of the OH group of native quercetin. We performed the synthesis by adjusting the equivalent molar ratio to attach the oleyl moiety to the 3-OH and 7-OH positions. After the reaction, a mixture of compounds with several oleyl moieties was obtained. The initial reaction products were isolated and assayed for tyrosinase and antioxidant activity. The final single compound was isolated and purified by open column chromatography, medium-pressure liquid chromatography (MPLC), and high-performance liquid chromatography (HPLC) (Appendix A), and the structure of the purified compound was confirmed by NMR spectroscopy (Appendix A) [14]. The yield of OQ was 11%.

### 2.2. Inhibition Kinetics

The effects of quercetin and OQ on the oxidation of L-3,4-dihydroxyphenylalanine (L-DOPA) were studied using the method described by Chen and Kubo [15]. The inhibition of tyrosinase by OQ was concentration-dependent, as shown in Figure 1. As the OQ concentration increased, the residual enzyme activity rapidly decreased, but it was not completely suppressed. The inhibition concentration leading to 50% activity loss (IC_50_) was estimated to be 0.0987 mM, which was the highest among the native quercetin and quercetin derivatives (Table 1). The plot of the residual enzyme activity versus the enzyme concentrations at different OQ concentrations produced a family of straight lines (Figure 1), which passed through the origin, indicating that the inhibition of the enzyme by quercetin was reversible. Increasing the inhibitor (OQ) concentration resulted in a decrease in the slope of the lines. The plot of 1/ν versus 1/[S] produced a series of straight lines, indicating that OQ is a competitive inhibitor. Thus, the kinetic behavior of mushroom tyrosinase during the oxidation of L-DOPA was depicted as a reversible competitive inhibition model, as follows (Scheme 2): Where, E (E_met_, E_deoxy_, and E_oxy_), S, I, and P denote the enzyme (three forms of the enzyme), substrate, inhibitor (quercetin and OQ), and product, respectively. EI and ES are the respective compounds.

The kinetic parameters of mushroom tyrosinase obtained from the Lineweaver-Burk plot showed that K_m_ was equal to 0.736 mM and V_max_ was equal to 30.2 U/min (8.1 μM/min). Further, the Lineweaver-Burk plot was linearly fitted (shown in Figure 1b, implying that OQ had a single inhibitory site or a single class of tyrosinase inhibitory sites [16]. The inhibition constant (K_i_) of OQ was calculated to be 0.232 mmol L^−1^ in the second plot of the apparent K_m_/V_m_ or 1/V_m_ versus OQ concentration (Table 1). As K_m_ and V_max_ are known quantities from measurements of the substrate reaction in the absence of the modifier at different substrate concentrations, the microscopic rate constants (k_+0_ and k − k_-0_) were easily determined according to the analysis of Tsou’s kinetic model [17] and are summarized in Table 1. Our data showed that the kinetic parameter values, except for V_m_, were similar for quercetin and OQ.

### 2.3. Intrinsic Fluorescence Quenching Analysis

The conformational change of tyrosinase has often been investigated using tryptophan fluorescence in tyrosinase molecules [19]. In the present study, the conformational alteration of the enzyme was evaluated by measuring the intrinsic fluorescence intensity of the enzyme before and after the addition of OQ [20]. As shown in Figure 2a, tyrosinase exhibited intense fluorescence emission with a peak at 380 nm. The OQ concentrations for curves 1–4 were 0, 0.6, 0.8, and 1.0 mM, respectively. Curve a is the fluorescence intensity of OQ at a concentration of 1.0 mM. The maximum fluorescence intensity decreased as the OQ concentration increased. The relative fluorescence intensity of tyrosinase was reduced to 50% when the OQ concentration was 0.8 mM (Figure 2b). These results indicate that OQ induced changes in enzyme conformation and decreases enzyme activity after binding to the enzyme molecule. To establish the interaction mechanism of OQ with tyrosinase, the Sternvolmer plot f was utilized, and it was confirmed that it was one type in the quenching process (Figure 2c). The bonding constant decreased with increasing temperature (Figure 2d).

### 2.4. Molecular Docking Study

Molecular docking aims to predict the ligand-protein binding structure by modeling the interaction between the ligand and protein at the atomic level. Using this method, it is possible to investigate the behavior of the ligand within the binding site of the target protein. The binding among quercetin and OQ, mushroom, and human tyrosinase is shown in Figure 3. In mushroom tyrosinase and human tyrosinase-related protein-1 (hTRP-1), there are hydrophobic pockets surrounded by Met 257, His 244, His 259, His 85, His 61, Val 283, His 263, Arg 267, Phe 264, Arg 74, Arg 362, His 215, Thr 391, His 392, Gln 390, His 381, and Arg 321 In these hydrophobic pockets, quercetin and OQ formed ligand conformations that inhibited enzyme activity. For quercetin, there was a difference in the binding model of mushroom tyrosinase and hTRP-1 (Figure 3a,e). For OQ, mushroom tyrosinase and human tyrosinase showed similar binding patterns (Figure 3c,g). The 3,4-dihydroxy-phenyl ring of OQ is located at the active site of tyrosinase, and the oleyl functional group contacts the surface. Yu et al. [21] performed molecular docking to confirm the synergistic effect of quercetin, cinnamic acid, and ferulic acid on mushroom tyrosinase and found that quercetin binds to the same active site as in the present study.

### 2.5. Cell Viability and Melanin Contents of Native Quercetin and OQ

The cytotoxicity of native quercetin was measured at concentrations of 5–30 µg/mL (17–99 µM) in B16F10 melanoma cells (Figure 4a). Quercetin showed very high toxic levels below 5 μg/mL (17 µM) after quercetin treatment at 1 µg/mL for 30 min and α-melanocyte-stimulating hormone (α-MSH) for 48 h (Figure 4c). OQ was treated at concentrations of 25–100 µg/mL (31–125 µM) in B16F10 melanoma cells (Figure 4b). OQ was measured to have lower toxicity then quercetin and did not show cytotoxicity even at a concentration of 100 µg/mL (125 µM). After 48 h of treatment with OQ 50–100 µg/mL (62–125 µM) and α-MSH 1 µg/mL, the melanin content decreased in a dose-dependent manner (Figure 4d,e).

### 2.6. Effet of OQ on Anti-Melanogenesis-Elated Proteinsin B16F10 Cell Line

Melanogenesis mainly depends on the regulation of melanogenic proteins such as tyrosinase, tyrosinase-related protein 1 (TRP-1), and tyrosinase-related protein 2 (TRP-2). Western blot analysis was performed to determine whether the inhibitory effects of OQ were related to the regulation of melanogenesis-related proteins. In general, microphthalmia-associated transcription factor (MITF) is a well-known master transcription factor of three major pigmentation enzymes: tyrosinase, TPR-1, and TRP-2 [22]. MITF regulates melanocyte differentiation and the transcription of melanogenic enzymes, such as tyrosinase, TRP-1, and TRP-2, to activate or regulate protein expression [22]. As shown in Figure 5a,b, the protein expression levels of tyrosinase, TRP-1, and TRP-2 were increased by the α-MSH treatment, whereas OQ led to a significant decrease in tyrosinase, TRP-1, and TRP-2 in B16F10 cell lines. Moreover, α-MSH-induced (1 µg/mL) MITF expression was decreased in a dose-dependent manner by treatment with OQ at 50–100 µg/mL (62–125 µM) (Figure 6a,b). Therefore, OQ inhibits melanogenesis by downregulating MITF signaling. To elucidate the mechanism underlying the melanogenesis of OQ, B16F10 cells were exposed to OQ (50–100 µg/mL (62–125 µM)) for the indicated incubation time, and the protein extracts were then analyzed by western blotting analysis. As shown in Figure 7, OQ preincubation inhibited α-MSH-induced phosphorylation of protein kinase and cAMP response element-binding protein (PKA/CREB). Therefore, the suppressive mechanism of OQ was related to the inhibition of PKA/CREB signaling.

### 2.7. Effect of OQ on Anti-Melanogenesis-Related Genes in B16F10 Cells

The effect of OQ on the expression of melanin production-related genes was assessed using mRNA expression analysis (Figure 8). The mRNA expression of all genes (tyrosinase, TRP-1, and MITF) was triggered by α-MSH, whereas OQ treatment resulted in significantly decreased mRNA expression (50–100 µg/mL [62–125 µM]). In particular, 100 µ/mL (125 µM) of OQ was more effective in decreasing mRNA expression than the other concentrations. OQ showed a more potent inhibitory activity than the positive control, arbutin.

### 2.8. Zebrafish Model Study

The whitening effect of quercetin and OQ was verified using AB line zebrafish. The control group without any compound treatment showed obvious pigmentation. 1-phenyl 2-thiourea (PTU) was used as a positive control, and a concentration of 30 µg/mL of PTU made the zebrafish transparent without interfering with the developmental process (Figure 9). Quercetin was treated within the non-toxic concentration range (2.5–5 µg/mL) and did not show a whitening effect at the total concentration tested. However, OQ showed obvious inhibitory effects on zebrafish pigmentation in a dose-dependent manner. In a certain concentration range (50–100 µg/mL), the inhibitory effects increased as the concentration increased. OQ had a significant effect on melanin synthesis in zebrafish (*p* ≤ 0.05).

## 3. Discussion

Previous studies have reported the isolation and purification of quercetin from plant extracts. However, limited research has been undertaken on the artificial synthesis of quercetin derivatives. Here, we attempted to synthesize a derivative mixture using quercetin and oleyl bromide by adjusting the molar ratio for the first time. We repeatedly succeeded in producing OQ using 1.5 equivalent conditions (Scheme 1). Quercetin has five hydroxyl groups; its B-ring comprises a catechol group and the 5-OH group of the A-ring forms an intramolecular hydrogen bond with the C-4 carbonyl group. It is well known that the hydroxyl group of quercetin has similar reactivity with the 3-position (3-OH), 7-position (7-OH), and 4′-position (4′-OH). These positions have high electron spin densities and acidities, but different steric structures [13,23,24,25]. Therefore, when a nucleophilic substitution reaction is performed using a brominated compound, substitution reactions occur simultaneously at positions 3, 7, and 4′. Unless the reaction conditions are specifically controlled, it is not easy to substitute one or two functional groups in the OH position, except for the 4′position. Previous studies have shown that the hydroxyl group of quercetin occurs gradually in the sequential order of 4′ > 7 > 3 > 3′ > 5 [26]. The catechol group on the B ring (3′-OH and 4′-OH) of quercetin binds to the active site of tyrosinase and acts as an inhibitor. Moreover, other studies have confirmed the alkylation reaction of the quercetin hydroxyl group was 7 > 4′> 3 > 3 > 5 [27]. In the present study, we purified the oleyl derivative of quercetin at positions 3 and 7 by screening for tyrosinase inhibition. Because the hydroxyl group of quercetin has a different reactivity when synthesized depending on the position, most quercetin synthesis has been performed at the desired position after protecting the -OH group. In a previous study, protection was carried out in 3-step, and n-butyl bromide, allyl chloride, cinnamyl chloride, and geranyl bromide were synthesized at the 3rd and 7th positions [28]. Al-Jabban et al. [29] confirmed that the cytotoxic efficacy varied depending on the location of the hydroxyl group synthesized in quercetin. The alkyl group was adjusted to three equivalents and reacted in one step. It was synthesized at positions 3,4′,7 of quercetin, and the derivatives were found to be more effective than quercetin in prostate cancer cells. Kim et al. [30] suggested that the longer the length of the alkyl chain, the better the antioxidant activity. In our study, synthesis was performed using a novel method that has not been used in previous studies. The monomers were fixed at the 3rd and 7th positions of the OH group of quercetin via molar equivalence control. Our findings suggest that a novel whitening agent can be developed by a selective substitution reaction using simple halide compounds such as oleyl bromide. In the future, it is necessary to evaluate the toxicity and whitening of the quercetin derivative synthesized in the form of trimers, and it will be of scientific significance to evaluate the derivative activity using a chloride-leaving group.

For theoretical studies, enzyme kinetics were performed to demonstrate the whitening activity of the new compound using an engineering system. In previous studies, it was reported that the K_I_ value (inhibition constant) and tyrosinase inhibitory concentration of the quercetin derivative with a catechol group in the B ring differed according to the molecular structure of the synthetic material. The tyrosinase inhibitory activity of quercetin was previously described from its ability to chelate copper in the binuclear active center of the enzyme [31]. Chen et al. [15] measured the kinetics of tyrosinase inhibition by quercetin, and was a competitive inhibitor of tyrosinase, with a K_I_ value of 0.0386 mM. Hence, quercetin and OQ inhibit the enzyme competitively, which is a competitive inhibition mechanism. The inhibition of tyrosinase by OQ is a slow and reversible reaction involving the activity of other enzymes, such as quercetin. The kinetic constants of OQ were consistent with those of quercetin, except for V_max_ (Table 1). However, there were some discrepancies between the kinetic values of quercetin from different sources. Controversial data have been published using the same raw material as quercetin from different sources [3]. Thus, the source and preparation method of quercetin must be described before concluding the IC_50_ and parameter values of the tyrosinase enzyme kinetics model. The pyrone moiety is responsible for the inhibitory activity of tyrosinase because it preferentially chelates copper in the enzyme, even if other moieties exist in the same molecule. In the present study, the measured K_I_ value of quercetin was 0.2459 mM. The K_I_ values could be different from those in previous studies because they are affected by the reaction time and substrate and enzyme concentrations. Previous studies have synthesized quercetin-7-oleate using quercetin and oleic acid, and their inhibition of tyrosinase was measured [18]. Quercetin-7-oleate is a competitive inhibitor of tyrosinase. To determine which quercetin derivative is most beneficial in promoting anti-melanogenesis effects, each hydroxy group of the quercetin skeleton was selectively protected.

Melanin is a polymorph and a multifunctional biopolymer. Melanin is synthesized in melanocytes, which are localized in the basal layer of the epidermis. Melanogenesis depends on the regulation of melanogenic proteins, such as tyrosinase, TRP-1, and TRP-2. Many signaling pathways produce melanin, and all signals eventually upregulate MITF. Activated MITF stimulates TRP-1, TRP-2, and tyrosinase expression; therefore, melanin is formed [21]. We investigated the inhibitory effect of OQ on melanogenesis and found that it downregulated tyrosinase, TRP-1, TRP-2, and MITF expression. The regulation mechanism of OQ might be the same as that of quercetin and should be further studied. While the present study focused on tyrosinase, other regulation pathways of melanin would be possible, such as feedback loop regulation and kinase inhibition. Several protein kinases (protein kinase C, Ca/calmodulin-dependent protein kinase, tyrosine kinase, and phosphatidylinositol-3-kinase) present in cells are involved in melanogenesis regulation [32]. The promotion of melanogenesis increases tyrosinase expression by factors that increase cyclic AMP or by post-translational modification of an already existing enzyme [33]. As evidence for the increase in tyrosinase expression, cyclic AMP has been reported to increase the m-RNA expression against tyrosinase [34]. Therefore, OQ reduces intracellular cyclic AMP concentration and tyrosinase via cyclic AMP-dependent protein kinase. To determine at which stage various protein kinases are involved in the signaling pathway of MSH, it is necessary to process PMA, a protein kinase C activator, and confirm it via a downregulation process in the future.

There has been controversy in previous studies over the range of toxicity and whitening activity in quercetin. The anti-melanogenesis effect of quercetin is unclear, and it remains to be determined whether quercetin induces an increase or decrease in melanin content. The effect might depend on the quercetin concentration, which should be tested both in vitro and in vivo. Regarding the controversy over the whitening activity of quercetin, Choi and Shin [3] investigated and reported the concentration range and toxicity range of whitening activity. When the quercetin concentration ranged from 10 to 20 µg/mL, melanin content was increased, and when quercetin concentration ranged from 20 to 50 µg/mL, melanin content decreased in cells. Therefore, quercetin showed strong toxicity even at concentrations below 10 µg/mL and no whitening effect at the concentration level; therefore, quercetin was not suitable as a safe whitening agent. In the present study, synthesis was conducted based on the quercetin structure to develop a new whitening material that was safe and useful when applied to the skin. OQ, the novel compound developed, showed a high whitening effect and did not show cytotoxicity even at concentrations below 100 µg/mL. Previous studies have confirmed the anti-melanogenic activity of β-arbutin, kojic acid, bis(4-hydroxybenzyl)sulfide, and inularin using a zebrafish model. The concentrations that resulted in reduced pigmentation levels compared to the control were 1000, 25, 10, and 10 µM [35,36]. There have been no studies on the anti-melanogenesis activity of zebrafish using quercetin and quercetin derivatives. In the present study, the whitening activity and toxic concentration of quercetin and quercetin derivatives were confirmed through in vitro and in vivo experiments, which were not found in previous studies. To quantitatively analyze tyrosinase inhibition activity, a three-dimensional melanoma cell culture technique for melanin quantification should be used in future research [37].

In some studies, it was found that tyrosinase activity in cell systems was enhanced; however, there was no effect on protein expression, resulting in the overexpression of tyrosinase due to quercetin treatment at concentrations of 1–20 µM [6]. Takekoshi et al. [7] showed an increase in melanin content at quercetin concentrations > 50 µM [7]. Moreover, tyrosinase and TRP-2 were overexpressed at quercetin concentrations of 5–160 µM, but there was no effect on TRP-1 at quercetin concentrations of 50–160 µM. Similarly, the same group showed that at 10 µM quercetin, melanin content increased and tyrosinase was overexpressed after 3 days [8]. The efficacy of quercetin derivatives as cosmetic ingredients is somewhat certain; however, further studies on the synthesis of various compound libraries and molecular signaling should be undertaken using suitable materials and methods.

## 4. Materials and Methods

### 4.1. Chemicals

Quercetin, tyrosinase (EC 1.14.18.1, 128 kDa), and l-DOPA were purchased from Sigma Aldrich (St Louis, MO, USA). Stock solutions of tyrosinase and L-dopa (5.0 × 10^−3^ mol L^−1^) were dissolved in 0.05 mol L^−1^ sodium phosphate buffer, pH 6.8. Oleyl bromide, Na_2_CO_3_, and dimethyl sulfoxide (DMSO) were purchased from Sigma Aldrich. Arbutin was purchased from TCI (Tokyo, Japan), and PTU and quercetin were purchased from Sigma Aldrich. PTU is a well-known tyrosinase inhibitor that is routinely used to inhibit pigment production in zebrafish. All other chemicals and reagents were of molecular biology grade and were commercially available. Antibodies against tyrosinase, TRP-1, TRP-2, MITF, PKA, p-PKA, CREB, and p-CREB were obtained from Santa Cruz Biotechnology (Santa Cruz, CA, USA). p-PKA and p-CREB antibodies were purchased from Cell Signaling Technology (Danvers, MA, USA). α-MSH, DMSO, sodium nitrite, and an antibody against β-actin were obtained from Sigma Chemicals (St. Louis, MO, USA). Horseradish peroxidase-conjugated goat anti-rabbit and anti-mouse antibodies were obtained from Invitrogen (Carlsbad, CA, USA).

### 4.2. Synthesis of 3,7-dioleylquercetin (OQ)

The synthesis of OQ from quercetin and oleyl bromide was as follows: OQ was prepared using the S_N_2 reaction. 3,7-dioleylquercetin: The quercetin derivatives were prepared by mixing quercetin (0.5000 g, 1.65 mmol), oleyl bromide (0.8223 g, 2.48 mmol), and sodium carbonate (0.2630 g, 2.48 mmol) in 15 mL of dimethyl sulfoxide at 25 °C for 30 h. Conversion of quercetin to many derivatives was 100%. The resultant solution was partitioned with ethyl acetate (EtOAc) and washed with a saturated aqueous solution of NH_4_Cl and NaCl. The organic layer was dried over anhydrous MgSO_4_ and then concentrated. Medium-pressure liquid chromatography was performed using PUMP 582 (Yamazen, Osaka, Japan) and PREP UV-10V (Yamazen) (Appendix A). The experiment was conducted under the following conditions: flow rate, 20 mL/min; UV detector, 254 nm; column, ULTRA PACK (20 mm × 200 mm, 30 µm) (Appendix A). The solvents used were EtOAc (A) and n-hexane (B). The gradient program of 4-methylphenethyl quercetin was (B) 90%, 0–5 min; (B) 40%, 5–55 min; (B) 40%, 55–65 min; (B) 0%, 65–70 min; (B) 100%, 70–80 min. The gradient program of 2-methylpropane quercetin was (B) 90%, 0–20 min; (B) 40%, 20–70 min; (B) 40%, 70–80 min; (B) 0%, 80–85 min; (B) 100%, 85–135 min. The gradient program of oleyl quercetin was (B) 100%, 0–10 min; (B) 90%, 10–50 min; (B) 80%, 50–90 min; (B) 70%, 90–105 min; (B) 50%, 105–120 min; (B) 0%, 120–125 min; (B) 0%, 125–135 min. OQ was isolated by preparative high-performance liquid chromatography with an X-Bridge Prep OBD C18 column (5.0 µm, 19 mm × 150 mm). Elution was performed with a linear gradient of methanol (0 min, 50/50; 30 min, 100/0; 100 min, 100/0) to obtain OQ as a yellowish powder with 11% yield. ^1^H and ^13^C NMR spectra were recorded in methanol-d4 using AVANCE III HD 400 MHz NMR (Bruker, Billerica, MA, USA) (Appendix A). Coupling constants were expressed in Hz and chemical shifts were expressed on a d (ppm) scale. ^1^H NMR (CD_3_OD): δ = 0.77 (m, 3H), 1.22 (m, 16H), 1.30 (m, 2H), 1.39 (m, 4H), 1.65 (m, 2H) 1.92 (m, 4H), 3.90 (dt, 2H, J = 48.00 Hz, J = 6.40 Hz), 5.23 (m, 2H), 6.20 (d, 1H, J = 2.00 Hz, H-8), 6.43 (d, 1H, J = 2.40 Hz, H-6), 6.79 (d, 1H, J = 8.40 Hz, H-5′), 7.42 (dd, 1H, J = 8.40 Hz, J = 2.00 Hz, H-6′) 7.50 (d, 1H, J = 2.40 Hz, H-2′). ^13^C NMR (CD_3_OD): δ = 13.05, 22.33, 25.60, 26.69, 26.73, 28.67, 28.75, 28.80, 28.95, 29.05, 29.21, 29.36, 29.45, 29.54, 31.67, 72.552, 92.12 (C-8), 97.80 (C-6), 105.31 (C-4a), 114.81 (C-2′), 115.40 (C-5′), 121.18 (C-6′), 121.63 (C-3), 129.41, 129.49, 137.44 (C-1′), 145.02 (C-2), 148.59 (C-3′), 156.91 (C-4′), 157.25 (C-5), 161.50 (C-8a), 165.22 (C-7), 178.83 (C-4).

### 4.3. In Vitro Tyrosinase Inhibition Assay and Enzyme Kinetics

Tyrosinase is well known as a key enzyme that determines the overall oxidative kinetics of L-tyrosine to 3,4-dihydroxy-L-phenylananine (DOPA) to DOPA-quinone. Therefore, in the present study, L-DOPA substrate was used, and experiments were performed at 0.5, 1.0, 1.5, and 2.0 mM. Tyrosinase inhibition activity was detected with a spectrophotometer (Shimadzu UV-2450, Shimadzu, Japan), and an IC_50_ quantitative assay of tyrosinase was performed according to the method described by Fan et al. [38]. In brief, aliquots of 20 µL of a solution composed of 1750 U/mL of mushroom tyrosinase (Sigma Aldrich) were added to 96-well microplates, and then 100 µL of pH 6.8 phosphate buffer solution and 60 µL of OQ (100 µg/mL, in 25% DMSO) was added.

The absorbance of the microplate wells was read using a microplate reader (Synergy HT, BIO-TEX, Winooski, VT, USA) at 510 nm (T0). Then, the microplates were incubated at 27 ± 1 °C for 30 min, and the absorbance was measured again (T1). An additional reaction period of 30 min at 30 ± 1 °C was performed, after which a new spectrophotometric reading was completed (T2). The inhibitory percentages at the two time points (T1 and T2) were obtained based on the following formula: IA% = (c − S)/c × 100, where IA% = inhibitory activity; C = negative control absorbance; S = sample or positive control absorbance (absorbance at time T1 or T2 minus the absorbance at time T0).

### 4.4. Quenching Study

The fluorescence assay of tyrosinase in the presence of OQ was measured according to the methodology described by Kim et al. [19] using a fluorescence spectrophotometer (Synergy HT, BIO-TEX, VT, Winooski, USA) [19]. Experimental conditions and some equations for data analysis (e.g., Stern–Volmer equation) followed the procedures described by Chai et al. [39].

Fluorescence quenching results from a change in processes, including dynamic and static quenching, and a combination of these processes. When the type is static, the apparent binding constant (*K_A_*) and the number of binding sites (*n*) can be estimated using the following equation:(1)$$\log[(F_{0}-F)/F]=\log K_{A}+n\;\log\left[Q\right]$$

The types of non-covalent interactions are known as multiple hydrogen bonds, hydrophobic forces, and van der Waals interactions.

If the temperature did not vary drastically, the enthalpy change (∆*H*^0^) is regarded as a constant, and its value and the value of entropy change (∆*S*^0^) calculated from the van’t Hoff equation:(2)$$\log K_{A}=-\frac{∆H^{0}}{2.303\;RT}+\frac{∆S^{0}}{2.303R}$$

The free energy change (∆*G*^0^) was determined from the following equation:(3)$$∆G^{0}=∆H^{0}-T\;∆S^{0}$$
where, *T* is the absolute temperature used in the experiments. KA is the binding constant at the corresponding temperature. *R* is the gas constant (8.314 J mol^−1^K^−1^).

### 4.5. Molecular Docking Procedure

Molecular docking was performed to confirm the binding site of mushroom tyrosinase and TRP-1 to quercetin and OQ using the Glide module in Schrodinger [40,41]. The X-ray crystal structures of tyrosinase (PDB ID: 2Y9X) and TRP-1 (PDB ID: 5M8O) were retrieved from the Protein Data Bank (http://www.rcsb.org (accessed on 10 October 2020)). The retrieved protein structures were processed using Protein Preparation Wizard in the Schrodinger package to remove the crystallographic water molecules, add hydrogen atoms, and assign protonated states and partial charges. The missing side chains and loops were built and refined using the Prime tool of the Schrodinger suite [42]. All protein residues were parameterized using the OPLS3e force field [43,44]. Finally, restrained minimization was performed until the converged average root mean square deviation of heavy atoms was 0.3 Å. Docking studies of quercetin and OQ with mushroom tyrosinase and TRP1 were performed using the Glide docking tool in the Schrodinger package. Docking grid boxes were generated considering the catalytic sites of mushroom tyrosinase and TRP-1. Quercetin and OQ were docked into the catalytic site of each protein using standard precision scoring modes. The 3D molecular structures of quercetin and OQ were minimized using the macromodel module of Schrodinger.

### 4.6. Cell Culture

B16F10 melanoma cells, a murine melanoma cell line, were purchased from the American Type Culture Collection (Rockville, MD, USA). Cells were maintained in Dulbecco’s modified Eagle’s medium (DMEM) supplemented with 10% fetal bovine serum (FBS), 50 units/mL penicillin, and 50 µg/mL streptomycin at 37 °C in a humidified atmosphere with 5% CO_2_ at 37 °C.

### 4.7. MTT Cell Viability Assay

The cytotoxicity proliferation assay was performed using the 3-(4,5-dimethylthiazol-2-yl)-2,5-diphenyltetrazolium bromide (MTT) assay. B16F10 cells were cultured at 1 × 10^4^ cells/cm^3^ in 6-well plates. After 24 h, the cells were treated with 62 and 125 µM OQ for 48 h. At the end of the incubation, 100 µL of MTT solution (1 mg/mL in DMEM) was added to each well. After incubation at 37 °C for 1 h, the medium was gently removed, and 400 µL of DMSO was added. The absorbance of each well was measured at 570 nm using a spectrophotometer.

### 4.8. Measurement of Melanin Content

Melanin content was determined according to the method described by Hosoi et al. [45]. B16F10 cells were cultured at 1 × 10^4^ cells/cm in 6-well plates. After 24 h, the cells were stimulated with α-MSH 1 µg/mL. Simultaneously, various concentrations of OQ (62 and 125 µM) were added for 48 h. After washing with phosphate-buffered saline (PBS), cells were harvested by trypsin treatment. The collected cells were dissolved in 100 µL of 1 N NaOH and measured at 405 nm using a spectrophotometer.

### 4.9. RNA Isolation and Reverse Transcription Polymerase Chanin Reaction (RT-PCR)

Total RNA was extracted using TRIzol reagent (Invitrogen) according to the manufacturer’s instructions. To obtain cDNA, total RNA (2 µg) was reverse-transcribed using an oligo primer (dT). The cDNA was amplified using a high-capacity cDNA synthesis kit (Bioneer, Daejeon, Korea) with a PCR machine (Bio-Rad, Hercules, CA, USA). Subsequently, PCR was performed using a PCR premix (Bioneer), and real-time RT-PCR was performed with STEP ONE (Applied Biosystems, Foster City, CA, USA) using a SYBR Green premix in accordance with the manufacturer’s instructions (Applied Biosystems). Primers were synthesized by Bioneer. The following primer sequences were used: mouse tyrosinase 5′-ATAACAGCTCCCACCAGTGC-3′ (sense) and 5′-CCCAGAAGCCAATGCACCTA-3′ (antisense); mouse MITF, 5′-CTGTACTCTGAGCAGCAGGTG-3′ (sense) and 5′-CCCGTCTCTGGAAACTTGATCG-3′ (antisense); mouse TRP-1 5′-AGACGCTGCACTGCTGGTC AAGCCTGTAGCCCACGTCGTA-3′ (sense) and 5′-GCTGCAGGAGCCTTCTTTCT-3′ (antisense). The expression of glyceraldehyde 3-phosphate dehydrogenase was used as an endogenous control for qRT-PCR experiments [46].

### 4.10. Zebrafish Study

Zebrafish were provided by the Shandong Academy of Sciences as an AB line, and the screening method was as follows [47]. Embryos were obtained from natural spawning that was induced in the morning by turning on the light. Embryos were collected within 30 min. After rinsing several times, the clean embryos were cultured at 28.5 °C. Approximately 100 zebrafish embryos treated with compounds (72 hpf) were collected in 1.5 mL EP tubes. After the zebrafish were washed twice with ice-cold saline water, a 10% full extract was prepared using a homogenizer. The tyrosinase extract was clarified by centrifugation at 10,000× *g* for 5 min at low temperature. A total of 100 µL of 1 mM L-DOPA and 480 µL of 1× PBS buffer were placed in a 1.5 mL EP tube to maintain the temperature at 37 °C. The mixed solution (290 µL) and tyrosinase extract (10 µL) were added to a 96-well plate. Absorbance was measured at 475 nm using a SpectraMax M5 microplate reader. A mixed solution (300 µL) was used as the control. The blank was removed from each absorbance value, and the final activity was expressed as the relative mean value in 5 min.

The compounds were dissolved in 0.1% DMSO and added to the embryo medium from 10 to 72 hpf. In the present study, we used 0.2 mmol/L PTU as the positive control group. We observed the pigmentation of zebrafish under a stereomicroscope, photographed at 55 hpf, and calculated the pigmentation area of each zebrafish using Image Pro Plus analysis software. All experiments were performed at least thrice with similar results.

### 4.11. Statistical Analysis

IBM SPSS version 26 was used for the statistical analysis of the data. One-way analysis of variance was used to assess the statistical significance of the differences among the treatment groups. For each statistically significant effect of treatment, Duncan’s multiple range test was used for comparisons between multiple group means. The data are expressed as mean ± standard deviation (S. D.) or standard error (S. E.).

## 5. Conclusions

Quercetin was chemically modified to prepare a novel whitening agent that produced OQ, resulting in reduced toxicity and enhanced anti-melanogenesis activity. The one-step synthesis step of the SN_2_ reaction is facile; however, the separation step should be optimized to increase the yield. Low toxicity and high activity were confirmed by various in vitro and in vivo methods. The novel OQ synthesized in the present study could be used in cosmetic products, especially during the oily phase, such as in creams, lotions, and serums, and the OQ concentration in cosmetics should be further studied. Human clinical trials should be performed to study the use of OQ ingredients in various cosmetic formulations.

## Data Availability

The data that support the findings of this study are available from the corresponding author upon reasonable request.

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
