# Peer review of "Novel Quercetin Derivative of 3,7-Dioleylquercetin Shows Less Toxicity and Highly Potent Tyrosinase Inhibition Activity"

_ijms, 2021, doi:10.3390/ijms22084264_

Round 1

Reviewer 1 Report

The manuscript needs to be complete rewritten to the academic English style. The construction of sentences did not respond the Englisch language, especially in the parts introduction, results and discussion

  We recommend that authors have their manuscripts checked by an English language native speaker before final approval of their submission;

Author Response

Question

The manuscript needs to be complete rewritten to the academic English style. The construction of sentences did not respond the Englisch language, especially in the parts introduction, results and discussion. We recommend that authors have their manuscripts checked by an English language native speaker before final approval of their submission;

Answer

Thank you for your comments. I have had the manuscript checked by an English editing service company and am resubmitting the revised version.

Reviewer 2 Report

            Inhibitors of the enzyme tyrosinase and the melanogenesis pathway are of great interest for cosmetic and therapeutic applications.  Here the authors synthesize a new quercetin derivative by direct alkylation at an unfavored pair of positions.  They fully characterize its interactions with mushroom tyrosinase, demonstrating competitive inhibition and a binding-induced conformational change.  The binding is further explored computationally by a docking study, extending the investigation to human analogs.   Next the authors test the compound versus relevant controls in two biological model systems, melanoma cell culture and zebrafish embryos.  In each case it shows low toxicity and efficacy.  Further, the authors trace the cellular behavior to decreased production of proteins in the melanin pathway, via inhibition of PKA/CREB signaling.  Overall, this manuscript covers a great deal of ground in exploring this new potential skin-whitening agent.  The experiments are thorough with the enzyme kinetics particularly well studied.  It will be of significant interest in melanin field and to the general readership of IJMS.

  1. The cell culture findings beg the question of whether the compound’s ability to down regulate the melanin pathway is a result of binding the downstream proteins such as tyrosinase, TPR-1, perhaps via a feedback loop, or is an independent activity, perhaps through kinase inhibition.   While the manuscript is tyrosinase-focused, including a brief discussion of quercetin and derivatives as kinase inhibitors would be a useful addition. 
  2. Maybe I am misunderstanding the way this is described, but I don’t see an inset in Figure 1c.  As it is presented in this manuscript, there is room for a Figure 1d to show this graph.
  3. Method 4.3 seems incomplete, it is not stated when L-dopa is introduced and at what concentrations.

Author Response

Reviewer_02

Question

  1. The cell culture findings beg the question of whether the compound’s ability to down regulate the melanin pathway is a result of binding the downstream proteins such as tyrosinase, TPR-1, perhaps via a feedback loop, or is an independent activity, perhaps through kinase inhibition. While the manuscript is tyrosinase-focused, including a brief discussion of quercetin and derivatives as kinase inhibitors, would be a useful

Answer

Thank you for your valuable comment. As you mentioned, this manuscript focuses on tyrosinase, but there are other possibilities. Therefore, the following discussion was added to the discussion section. While this study focused on tyrosinase, other regulation pathways of melanin would be possible, such as feedback loop regulation and kinase inhibition. Several protein kinases (protein kinase C, Ca/calmodulin-dependent protein kinase, tyrosine kinase, and phosphatidylinositol-3-kinase) present in cells are involved in melanogenesis regulation [32]. The promotion of melanogenesis increases tyrosinase expression by factors that increase cyclic AMP or by post-translational modification of an already existing enzyme [33]. As evidence for the increase in tyrosinase expression, cyclic AMP has been reported to increase m-RNA expression against tyrosinase [34]. Therefore, OQ reduces intracellular cyclic AMP concentration and tyrosinase via cyclic AMP-dependent protein kinase. To determine at which stage various protein kinases are involved in the signaling pathway of MSH, it is necessary to process PMA, a protein kinase C activator, and it via a downregulation process in the future.

References

Fuller, B.B.; Lunsford, J.B.; Iman, D.S. Alpha-melanocyte-stimulating hormone regulation of tyrosinase in Cloudman S-91 mouse melanoma cell cultures. J. Biol. Chem. 1987, 262, 4024-4033.

Park, H.Y.; Russakovsky, V.; Ao, Y.; Fernandez, E.; Gilchrest, B.A. α-melanocyte stimulating hormone-induced pigmentation is blocked by depletion of protein kinase C. Exp. Cell Res. 1996. 227, 70-79.

Kuzumaki, T.; Matsuda, A.; Wakamatsu, K.; Ito, S.; Ishikawa, K. Eumelanin biosynthesis is regulated by coordinate expression of tyrosinase and tyrosinase-related protein-1 genes. Exp. Cell Res. 1993, 207, 33-40.

Question

  1. Maybe I am misunderstanding the way this is described, but I don’t see an inset in Figure 1c. As it is presented in this manuscript, there is room for a Figure 1d to show this graph.

Answer

Thank you for your comment. We did not perform other kinetic experiments; therefore. Figure 1c is a Lineweaver-Burk plot graph, and its contents are described starting from line 116.

Question

  1. Method 4.3 seems incomplete, it is not stated when L-dopa is introduced and at what concentrations.

Answer

Thank you for your comment. Tyrosinase is well known to be a key enzyme determining the overall oxidative In this study, L-DOPA substrate was used and experiments were performed at 0.5, 1.0, 1.5, and 2.0 mM. This information has been included in the revised manuscript.